# Influence of Agronomic Practice on Total Phenols, Carotenoids, Chlorophylls Content, and Biological Activities in Dry Herbs Water Macerates

**DOI:** 10.3390/molecules26041047

**Published:** 2021-02-17

**Authors:** Kalina Sikorska-Zimny, Paweł Lisiecki, Weronika Gonciarz, Magdalena Szemraj, Maja Ambroziak, Olga Suska, Oliwia Turkot, Małgorzata Stanowska, Krzysztof P. Rutkowski, Magdalena Chmiela, Wojciech Mielicki

**Affiliations:** 1Skierniewice, Fruit and Vegetables Storage and Processing Department, Division of Fruit and Vegetable Storage and Postharvest Physiology, Research Institute of Horticulture, Pomologiczna 13a Street, 96-100 Skierniewice, Poland; krzysztof.rutkowski@inhort.pl; 2Stefan Batory State University, Batorego 64c Street, 96-100 Skierniewice, Poland; pawel.lisiecki@umed.lodz.pl (P.L.); majaambroziak98@gmail.com (M.A.); olgasuska98@gmail.com (O.S.); oliwia4811@wp.pl (O.T.); gosia.stanowska@op.pl (M.S.); wojciech.mielicki@umed.lodz.pl (W.M.); 3Department of Pharmaceutical Microbiology and Microbiological Diagnostics, The Medical University of Łódź, Pomorska 137 Street, 90-235 Lodz, Poland; magdalena.szemraj@umed.lodz.pl; 4Department of Immunology and Infectious Biology, Faculty of Biology and Environment Protection, The University of Łódź, Banacha 12/16 Street, 90-237 Lodz, Poland; weronika.gonciarz@biol.uni.lodz.pl (W.G.); magdalena.chmiela@biol.uni.lodz.pl (M.C.); 5Department of Pharmaceutical Biochemistry and Molecular Diagnostics, The Medical University of Łódź, Muszyńskiego 1 Street, 90-151 Lodz, Poland

**Keywords:** thyme, oregano, dry herbs, polyphenols, chlorophyll, carotenoids, microbial, cytotoxicity

## Abstract

Oregano (*Origanum vulgare* L.) and thyme (*Thymus vulgaris* L.) have long been known for their organoleptic properties. Both plants are widely used in cuisine worldwide in fresh and dried form and as a pharmaceutical raw material. The study aimed to assess if the type of cultivation influenced chosen chemical parameters (total polyphenols by Folin-Ciocalteu method; carotenoids and chlorophyll content by Lichtenthaler method), antimicrobial activity (with chosen reference microbial strains) and shaped cytotoxicity (with L929 mouse fibroblasts cell line) in water macerates of dry oregano and thyme. Polyphenols content and antimicrobial activity were higher in water macerates obtained from conventional cultivation (independently from herb species), unlike the pigments in a higher amount in macerates from organic herbs cultivation. Among all tested macerates stronger antimicrobial properties (effective in inhibiting the growth of *Pseudomonas aeruginosa, Bacillus cereus* and *Salmonella enteritidis*) and higher cytotoxicity (abilities to diminish the growth of L929 fibroblasts cytotoxicity) characterized the conventionally cultivated thyme macerate.

## 1. Introduction

Oregano (*Origanum vulgare* L., also known as wild marjoram), and thyme (*Thymus vulgaris* L.) are popular and widely used in cuisines and the cosmetic industry. Oregano and thyme are native species of the Mediterranean region [1,2] although they became popular herbs in the Baltic Sea region as well [3,4,5]. Due to their chemical composition, both herbs are willingly used in the cuisines and medicine/herbal medicine and cosmetics industries.

The properties of the oregano are connected not only with its flavor and scent, but this plant also supports the digestive system (secretion of gastric juices and bile), respiratory system (helps with catarrh of the upper respiratory tract), has a relaxing and expectorant effect, but also has antifungal and bactericidal properties [1,3]. Some of these pro-health properties are the result of the antioxidant effect of oregano [6]. This effect is related to the presence of numbers of compounds such as phenolic acids, flavonoids, and essential oils like sabinene, carvacrol, geranial, terpineol, linalool or cymene [1]. The concentration of these compounds is shaped not only by the variety and the environmental conditions of the plant during growth (greater exposure to sun, rain, the season of picking the plant) but also by the type of cultivation (organic, conventional), the geographical position but also by the method of oil extraction [6]. The beneficial antioxidant effect is a derivative of the (variable) composition, and therefore these properties will also be influenced by the factors mentioned. Antimicrobial activity of oregano is connected with compounds like carvacrol, thymol, γ-terpinene, and *p*-cymene [1], Faleiro et al. estimated the concentration as 33% for thymol, 26% for γ-terpinene and 11% for *p*-cymene [7].

Thyme in the cuisine is used as a spice for meat, potato, and mushroom dishes. The healing properties of thyme focus on the respiratory system’s effect; hence thyme is used in cough syrups, expectorants, and soothing throat irritations, in the form of ointments, drops used in the case of upper respiratory tract catarrh [8]. Thyme is also used in dental prophylaxis as a toothpaste component and liquids for teeth cleaning and oral care (also in endodontics, during disinfecting root canals) [9]. Thyme, similar to oregano, is used against digestive problems and has antibacterial, antifungal, and antiviral properties [6]. These effects are, inter alia, a derivative of the presence of thymol and carvacrol in thyme. These antioxidant compounds (phenolic derivatives) are present in thyme in large amounts: about 18–80% and 1–20% (for thymol and carvacrol, respectively) [10,11]. Two of these phenolic compounds (carvacrol and thymol) have strong antimicrobial activity against both bacteria and fungi, therefore, are considered to be useful as future fighting agents against drug-resistant microbial [12]. Detailed examination pointed on carvacrol with high anti-*Listeria monocytogenes* properties [13], experiments conducted by Rota et al. showed that carvacrol and thymol had a high inhibitory effect against the Gram-positive *L. monocytogenes* and *S. aureus,* the Gram-negative *S. enteritidis, E. coli O157:H7, Yersinia enterocolitica,* and *Shigella flexneri* [14].

Different factors have been demonstrated to influence the content of the biologically active compounds, such as technological process, cultivation method, climatic conditions or variety. Among them technological processes such as drying increase the herbs’ availability on the market throughout the year, thereby not tied to climatic and cultivation conditions [15]. It enables transport over long distances, thus increasing the availability of these herbs on the market. Moreover, drying, by limiting the water content, helps to inhibit the growth of microorganisms on the product [16]. It needs to be remembered that the drying process changes the chemical composition of plants [15].

Other factors that influence the content of the biologically active compounds in plants are cultivation method, climatic conditions, variety, therefor content of, e.g., polyphenols may differ in published articles [17].

As mentioned before, oregano and thyme possess antioxidant properties connected with the presence of polyphenols compounds like phenolics: acids, diterpenes; flavonoids and volatile oils [18]. These broad group of secondary metabolites of plants are present in flowers, fruits, leaves, stems, roots, and tubers. The antioxidant properties of polyphenols are connected with the ability to scavenge free radicals, counteract lipid peroxidation, and reduced enzymes’ activity [17]. Polyphenols are sensitive for high temperatures; therefore, the drying process reduces their plant tissue content [19]. Consumers’ important factors during shopping are the color of dried spices, preferably intensive green (most similar to fresh plant) and less popular discolored in grey shades [20]. The color of a dried green plant is related to the content of chlorophyll and carotenoids. These pigments decompose during the technological procedures of drying herbs; therefore, except for visual evaluation of herbs color, the chemical analysis of chlorophylls and carotenoid content is useful.

The cultivation method influences all the features mentioned above. They compare organic and conventional method points on differentials in the usage of synthetic substances during and before plant growth. Organic cultivation method prohibits the use of synthetic plant protection products and fertilizers, which are substituted with pheromone traps, plant extracts and sticky boards (as a substitute of synthetic plant protection products) or manure (of different origin) and compost (as a substitute of synthetic fertilizers) [21,22]. Organically cultivated herbs are expected to be richer in bioactive compound due to elicitation phenomenon: “this process induces a defense response in a plant and leads to several biochemical processes resulting from which numerous groups of secondary metabolites are produced” [23].

The possibility to use both spices in cuisines is connected with their usage in the water solution. Therefore, important is to conduct an experiment with water macerates (of oregano and thyme) and examine their chemical and biological properties to evaluate the pro-healthy values.

The authors focused on determining the content of chosen compounds in water macerates of oregano and thyme, obtained from organic and conventional cultivation, its cytotoxicity, and the assessment of their impact on microorganisms’ development.

## 2. Results

### 2.1. Chemical Composition

#### 2.1.1. Polyphenols

The highest content of total polyphenols was determined in conventionally cultivated thyme (0.42 mg GAE/mL sample) and the lowest in “organic” oregano (0.20 mg GAE/mL sample). Total polyphenols content in water macerates for both herbs were higher in conventional cultivations of oregano and thyme; significant differences were obtained in “organic” oregano comparing to other samples. There were no significant differences among the thyme samples (Table 1).

#### 2.1.2. Chlorophyll and Carotenoids

Oregano had a higher content of chlorophyll (a and b) than thyme. The highest chlorophyll a and b content was determined in “organic” oregano (7.81 μg/mL and 12.23 μg/mL of macerates, for chlorophyll a and b, respectively), the lowest in “conventional” thyme (2.96 μg/mL and 4.56 μg/mL of macerates, for chlorophyll a and b, respectively). Chlorophylls (a and b) were significantly higher in organically cultivated herbs (for both species).

Carotenoids were in higher amount in macerates of organically cultivated herbs (2.07 and 1.67 μg/mL for oregano and thyme, respectively) than a conventional one (1.03 and 0.76 μg/mL for oregano and thyme, respectively).

The determined results (higher content of pigments in organic herbs) were not noticeable during the preparation (Figure 1 and Figure 2) of macerates, where colors of organic and conventional cultivated, dry plants were similar.

### 2.2. Antimicrobial Activity

The effects of the water macerates of both herbs on tested standard microbial are presented in Table 2. Four strains of Gram-negative bacteria and five strains of Gram-positive bacteria were used in the research. The experiment was repeated triplicate, and the results were consistent.

The macerates of both herbs obtained from organic cultivation did not exhibit antibacterial activity against standard strains of Gram-negative bacteria MIC/MBC > 50 mg/mL (Table 2). Oregano macerate obtained from conventional cultivation exhibited weak activity against *E. coli* and *Salmonella* Enteritidis (MIC/MBC = 50 mg/mL) and moderate activity against *P. aeruginosa* with MIC/MBC= 12.5 mg/mL. Thyme macerate received from conventional cultivation had been effective in inhibiting the growth of *S. enteritidis* (MIC/MBC = 12.5 mg/mL). The highest sensitivity to the conventional thyme macerate showed the tested strains of *P. aeruginosa* (MIC/MBC = 0.626 mg/mL).

Both macerates obtained from organic herbs were less active in inhibiting the growth of all the tested strains of Gram-positive bacteria, excluding *L. monocytogenes*, compare to conventional macerates (Table 2). The bactericidal effect of thyme conventional macerate against *S. aureus* was achieved at the concentration of 12.5 mg/mL, against *E. faecalis* and *B. subtilis* at 25 mg/mL. Thyme organic macerate was active against *B. subtilis* at concentration 25 mg/mL and did not inhibit the growth of *S. aureus* and *E. faecalis* (MIC/MBC > 50 mg/mL). The thyme and oregano macerates obtained from herbs cultivation in organic and conventional condition demonstrated weak activity against *L. monocytogenes* with MIC/MBC values, 50 mg/mL, and 25 mg/mL, respectively.

The “organic” oregano macerate was not active against *E. faecalis* (MIC/MBC > 50 mg/mL), but oregano conventional macerate was effective at concentration 25 mg/mL. The highest inhibitory effect on the tested strains of *S. aureus* and *B. subtilis* showed the oregano conventional macerate, resulting in MIC/MBC 3.125 mg/mL and 6.25 mg/mL, respectively (Table 2). The oregano organic macerate demonstrated activity against *S. aureus* and *B. subtilis* in higher concentrations with MIC/MBC 6.25 mg/mL and 25 mg/mL, respectively.

The highest sensitivity of conventional oregano and thyme macerates demonstrated Bacillus cereus’ strain, resulting in MIC/MBC 3.125 mg/mL and 6.25 mg/mL, respectively. The thyme and oregano organic macerates showed the same effect at twofold higher concentrations, 6.25 mg/mL, and 12.5 mg/mL, respectively.

The macerates’ antifungal activity was tested against two fungal strains—*C. albicans* (yeast fungus) and *A. brasiliensis* (mould fungus). None of the tested macerates and those obtained from herbs cultivation in organic and conventional conditions did not inhibit the fungal strains’ growth ((MIC/MBC > 50 mg/mL).

The tested macerates’ antibacterial activity was lower than the activity of gentamicin or fluconazole used as the antimicrobial reference standard.

The “conventional” thyme in the highest concentration of 50; 25 mg/mL and 12.5 significantly diminished the growth of L929 fibroblasts (43–71% of dead cells) (Figure 3), while thyme “organic” did not show cytotoxic activity at any of the tested concentrations (Figure 3). Moreover, “organic” thyme significantly increased the ability of L929 cells to reduce MTT within the range 5–50 mg/mL compare to extract isolated from thyme “conventional” (Figure 3).

Oregano “conventional” has a stronger inhibitory effect on L929 cell metabolic activity at range concentrations 10–50 mg/mL (40–60% of dead cells), while thyme “organic” in the highest concentration 50 and 25 mg/mL significantly diminished the growth of L929 fibroblasts (44–47% of dead cells) (Figure 3). Furthermore, oregano “organic” significantly increased the ability of L929 cells to reduce MTT within the range 50–2.5 mg/mL compare to extract isolated from oregano “conventional” (Figure 3).

## 3. Discussion

According to the authors’ best knowledge, there was no conducted analysis of organic and conventional cultivated oregano and thyme water macerates (not infusions). Available in literature are results over chemical analysis of herbs infusion dedicated as beverages (melissa, mint) or macerates based on other solution (methanol, ethanol), mostly used as a solvent in the method of extraction or infusions (pouring with hot water).

In conducted analysis higher polyphenol content was determined in thyme than in oregano that stands according to values obtained by Dragland et al. which determined 45 mmol/100 g and 63.7 mmol/100 g for oregano and thyme respectively in commercial cultivation [24]. Moreover, Vallverdú-Queralt et al. determined total polyphenols in dried oregano and thyme at the level of 2.23 and 3.36 mg GAE/g DW for oregano and thyme, respectively [25]. Jałoszyński determined polyphenols in dry oregano at the level: 63.96–168.87 mg GAE/100 g, where the lower content is similar to obtained in presented work (calculate on DM herbs) [19]. Taghipour et al. determined thyme’s polyphenols at a level of 30 mg GAE/g DM that is similar to values obtained in presented work [26]. It also needs to be highlighted that phenolic losses during drying of herbs can reach 50% (dependently of temperature and drying method). Generally, the compound degrades at a higher level with higher temperature applied [19].

Although determined higher polyphenols content in macerates for conventional cultivation stands against the work of Matłok et al., who determined the much higher bioactive potential of dry oregano cultivated with organic fertilizers comparing to conventional cultivation [27]. However, Lv et al. compared peppermint and cinnamon cultivated organically and conventionally and determined in both herbs higher total phenolic content in conventionally cultivated herbs [28].

Still, there is no clear explanation of higher polyphenol content in a specific type of plant cultivation (sometimes higher in organic, sometimes in conventional). Some authors divide polyphenols (into soluble and hydrolysable), where soluble are more stable during processing (cooking) [29], although there are still no data available about polyphenols profiles in dry herbs, together with water macerates. Moreover, the water content of fresh plant might decrease polyphenols content, since the drying process of high-water content must undergo a more extended drying time procedure, or higher temperature must be applied. In any case, both factors are reducing polyphenols content in the final product.

Chlorophylls play an important role in the photosynthetic membranes, where they are present usually in a ratio of 3:1 [30], although this content can be shaped by the growth conditions and environmental factors [31,32]. Chlorophyll a and b differ in a side chain’s composition, which for chlorophyll a its methyl group and chlorophyll b its aldehyde group.

Matłok et al. pointed on a strict correlation between chlorophylls content and carotenoids in foliar plants due to general biosynthetic pathways in the plants’ chloroplasts [27]. Therefore, the higher content of pigments involved in photosynthesis is related to higher carotenoid content (beta carotene) [21].

In the experiment over water macerates of herbs from organic and conventional cultivation, higher content of chlorophylls and carotenoids were determined in organic cultivation, which is opposite to Hallmann et al. [21]. Although Onofrei et al. pointed on a higher content of pigments presents in organically cultivated thyme and oregano [33]. It needs to be added that Hallmann et al. refer to lower content only of beta-carotene, and in the conducted experiment, the authors were determining the total carotenoids content [21]. Simultaneously Halmann et al. [21] pointed to the fact of a higher content of lutein and zeaxanthin (belonging to carotenoids) in organically cultivated herbs comparing to conventional one and in case of thyme major carotenoid is zeaxanthin [34].

Onofrei et al. explained the higher content of pigments in herbs cultivated with organic fertilizers by the possible improvement in the availability of nutrients caused by these fertilizers, and thus enhancing the metabolic pathways (e.g., photosynthesis) synthesis of several plant secondary compounds [33]. According to Skubij and Dzida, over the influence of organic fertilizers on the plants’ chemical compounds [35]. The other factor influencing carotenoid content is UV radiation that also increases antioxidant content [21,27].

Manukyan determined higher content of chlorophyll a and b in thyme at a level of 0.03–0.052 mg/g and 0.13–0.21 mg/g, (for chlorophyll a and b, respectively) [34]. Tzima et al. obtained the lower results: 88.6 mg/100 g DM for chlorophyll a, and 24.00 mg/100 g DM for chlorophyll b [36].

Kulbat-Warycha et al. determined similar to presented in work content of chlorophyll a in oregano but lower chlorophyll b and higher of total carotenoids (0.009 g/L; 0.002 g/L and 0.004 g/L FW for chlorophyll a, b, and carotenoids, respectively) [37].

Antimicrobial activity of water macerates was higher in conventional cultivated herbs. The obtained results go along with higher polyphenols content determined in these macerates. The highest antimicrobial activity (of “conventional” thyme) was determined against *Bacillus cereus*, which is especially important in dry herbs. These products may have a high number of bacteria and molds, which in favorable conditions, can quickly develop [38]. Research of Fogele et al. and Boer et al. pointed on a high level of *Bacillus cereus* determined in dry spices [38,39]. Boer et al. determined antimicrobial activity of 2% water macerates and pointed of weak or non-antimicrobial abilities of most of examined herbs solutions (rosemary, basil, ginger, sage) [38]. Other investigation showed that 20% cold water oregano extract was effective against the strains of *P. aeruginasa* and *K. pneumoniae* [40]. Our study demonstrated the highest antibacterial effect of thyme water macerate obtained from conventional cultivation against the strains of *P. aeruginosa*. This opportunistic human pathogen was found to contaminate herbal plants and spices caused by the unsafe collection, drying, preparation or storage [41].

Quadir et al. examined antimicrobial activity of chosen herbal extracts, but they have not confirmed any activity of thyme extract against *B. subtilis*, explanation to this fact may be connected with used solution concentrate (1:10) [42]. Our investigation shows that “conventional” oregano macerate exhibited moderate activity against *B. subtilis*.

Our study demonstrated that water macerates of oregano obtained from the conventional method of cultivation had been effective in inhibiting the growth of *S. aureus*. Other studies also pointed to a high activity oregano water extract against this microorganism [43]. *S. aureus* is most commonly isolated from dry herbs [41].

Research is connecting herbs’ antimicrobial ability with the presence of phenols that can precipitate proteins (or react with cells by sulfhydryl groups of proteins causing unavailability of the substrate) and inhibit microorganisms’ enzymes [44,45].

Plant extracts including thyme and oregano have been used in the traditional medicine for the treatment of several respiratory diseases like asthma and bronchitis [46] as well as other pathologic processes, thanks to several properties such as antiseptic, antispasmodic, antitussive, antimicrobial, antifungal, antioxidative, and antiviral [47,48]. Our study showed that extract isolated from thyme “organic” does not show cytotoxic activity according to ISO 10993-5 in all concentrations used in this study (1–50 mg/mL) whereas the activity of extract isolated from thyme “conventional” was below the cytotoxicity norm in the range 12.5–50 mg/mL. Similar results were observed in the study by de Oliveira et al. [49]. They showed that the *T. vulgaris* extract at 25, 50 and 100 mg/mL provided cell viability above 50% to RAW 264.7, FMM-1, MCF−7 and HeLa cell line [49]. New plant extracts are also tested for antioxidant activity in conjunction with the modulation of the inflammatory response. For example, Loizzo et al. examined different oregano’s oils varieties inhibition of NO production in the murine monocytic macrophage cell line RAW 264.7, obtaining an IC50 value of 66.4 μg/mL and >200 μg/mL for *O. ehrenbergii* and *O. syriacum*, respectively [50]. The antioxidant activity of our preparations with the use of gastric epithelial cells is planned to be tested in the context of minimizing the risk of epithelial damage and the development of an inflammatory reaction. Plant preparations with a cytotoxic effect against cancer cells are extremely valuable. Jamali et al. show that thymol induced toxicity, apoptosis, and cell cycle arrest in MDA-MB231 BC cells [51]. Our study shows that oregano extracts have strong cytotoxic activity against L929 fibroblasts. Oregano extracts: ethyl acetate and ethanol extracts from leaves also have reported the cytotoxicity activity against human breast cancer cells (MCF7) [52]. On the other hand, carvacrol, the major component of oregano, showed antimutagenic activity, which seems to be mainly linked to the induction of mitochondrial dysfunction [53,54].

## 4. Materials and Methods

### 4.1. Macerates Preparation

Dried oregano and thyme were bought in a local shop. Both types of herbs (organic and conventional cultivated) were obtained from the same manufacturer. Thyme and oregano (conventional and organic, separately) were soaked with distilled water (1:5) and left for five days for the maceration process. The extracts were filtered through a cellulose filter (fine pore, 0.45 µm) and then subjected to further analysis.

### 4.2. Chemical Analysis

#### 4.2.1. The Content of Polyphenols

The Folin-Ciocalteu assay was carried out according to the method described by Singleton and Rossi (1965) [55]. The results were expressed as mg/mL of the sample as gallic acid. Measurements were conducted in six replicates; absorbance was measured at 750 nm against the blank sample.

#### 4.2.2. The Content of Chlorophylls and Carotenoids

Carotenoids and chlorophyll a and b were determined according to Lichtenthaler (1983) spectrophotometric method, with the extraction of 80% acetone and absorbance measured at 470, 646, 663 nm [30].

### 4.3. Microbial Analysis

#### 4.3.1. Microbial Strains and Culture Conditions

Reference microbial strains were obtained from the American Type Culture Collection (ATCC), including *Staphylococcus aureus* ATCC 25923, *Enterococcus faecalis* ATCC 29212, *Bacillus subtilis* ATCC 6635, *Escherichia coli* ATCC 25922, *Pseudomonas aeruginosa* ATCC 27833 and *Shigella flexneri* ATCC 12022. The *Listeria monocytogenes* PCM 2191 and *Bacillus cereus* PCM 1948 strains were taken from the Polish Collection of Microorganisms (PCM). One bacterial strain *Salmonella* Enteritidis ZMF 279 was derived from the collection of the Department of Pharmaceutical Microbiology and Diagnostic Microbiology, Medical University of Lodz. Two fungal strains were also used: *Candida albicans* ATCC 10231 and *Aspergillus brasiliensis* ATCC 16404. All tested microorganisms were stored at –80 °C in 15% glycerol stocks. Before the investigation, the bacterial strains were transferred to Mueller-Hinton agar medium (Oxoid, Thermo Fisher Scientific, Waltham, MA, USA) and cultured overnight at 37 °C. Fungal strains were transferred on Sabouraud agar medium (Oxoid, Thermo Fisher Scientific, Waltham, MA, USA) and cultured for two days at 30 °C.

#### 4.3.2. Antimicrobial Assay

Before the investigation, the extracts were concentrated to 100 mg/mL by lyophilization. The antimicrobial activity of extracts was assessed according to their minimum inhibitory concentrations (MIC), and minimum bactericidal/fungicidal concentrations (MBC) expressed in mg/mL. According to the European Committee on Antimicrobial Susceptibility recommendations, antibacterial and antifungal activities were determined using the broth microdilution method [56]. The Mueller-Hinton broth (pH~7.2) (Oxoid, Thermo Fisher Scientific, Waltham, MA, USA) was used for bacteria. Liquid medium RPMI-1640 (w/o red phenol, pH~7.2) (Sigma-Aldrich, Darmstad, Germany) was used for the fungal strains. Two-fold series dilutions of extract in the growth medium were performed in 96-well sterile microtiter plates (Kartell Labware, Noviglio, Italy) in concentrations ranging from 50 to 0.09 mg/mL. The MIC values were defined as the lowest extract concentrations with no bacterial growth after the incubation. The MBCs were determined by seeding 5 μL from all clear MIC wells onto Mueller-Hinton agar plates (Oxoid, Thermo Fisher Scientific, Waltham, MA, USA) (bacterial strains) or Sabouraud agar medium (Oxoid, Thermo Fisher Scientific, Waltham, MA, USA) (fungal strains). MBC was defined as the lowest concentration that killed 99.9% of the final inocula after 24 h incubation at 37 °C (bacterial strains) or 48 h at 30 °C (fungal strains). The antimicrobial tests were performed in triplicate. Gentamicin and fluconazole were used as an antimicrobial reference standard.

### 4.4. Cytotoxicity Studies

#### 4.4.1. In Vitro Cell Culture

According to the ISO (International Organization for Standardization, 2009) norm 10993-5 (Biological evaluation of medical devices—Part 5: Tests for in vitro cytotoxicity), testing of cytotoxicity was performed using L929 mouse fibroblasts (LGC Standards, Middlesex, UK). The cells were maintained under standard conditions (37 °C, 5% CO_2_) in compelled culture medium (cRPMI-1640 medium supplemented with 10% fetal bovine serum (FBS) and antibiotics: 100 U/mL penicillin and 100 μg/mL streptomycin) Sigma-Aldrich (Darmstad, Germany) as previously described [57]. Before being used in the cytotoxicity assay, the cells’ viability was assessed by excluding trypan blue dye and was in the range of 93–95%.

#### 4.4.2. Measurements of Cellular Metabolic Activity and Global Growth Inhibition

The metabolic activity of the L929 cells was tested after application of water extracts obtained from dried plants. Cells in culture medium were seeded in 96-well plates (2 × 10^5^ cells/well) for 24 h at 37 °C, 5% CO_2_. The tested extracts were diluted in cRPMI-1640 medium in concentrations of 50; 25; 12.5; 10; 5; 2.5 and 1 mg/mL, added to the cells (100 μL/well), and incubated under standard conditions for 24 h. Following incubation, the cell monolayers were carefully screened using light microscopy, as recommended by ISO norm 10993-5, to evaluate cell morphology. Cell metabolic activity was estimated by measuring the ability of cells to reduce MTT [(3-(4,5-dimethylthiazol-2-yl)-2,5-diphenyltetrazolium bromide)], which is one of the tests recommended by the Food and Drug Administration (FDA) and ISO as previously described [57].

### 4.5. Statistical Analysis

Statistical analysis over chemical parameters was performed with Duncan’s test (*p* = 0.05). Obtained results are presented results are shown as the mean value of six replications. The statistical significance of the cytotoxicity results was determined by Kruskal-Wallis test (*p* < 0.05). Data are presented as mean values ± SD. For statistical analysis, the STATISTICA 12 and 13 PL software was used (Stat Soft, Kraków, Poland).

## 5. Conclusions

Many commonly used antibiotics are becoming useless due to the increasing resistance of pathogens. For this reason, there is an urgent need to search for new antimicrobial, anti-inflammatory and pro-regenerative drugs with high biocompatibility and targeted activity. Plant biocomponents with well-characterized properties and good biocompatibility are good candidates for developing new drugs, medicinal food, or dietary supplements. A significant impact on such formulations expected therapeutic value is the plant’s breeding conditions, limiting the content of potentially toxic compounds, or determining an active biological substance’s content.

Water macerates of oregano and thyme are a valuable solution. Dependently, they can be a good source of polyphenols and have good antimicrobial abilities (conventional) and are rich in pigments (organic). It is worth pointing that “conventional” thyme macerate exhibited strong activity against the strains of *P. aeruginosa*. This opportunistic pathogen alleged to cause gastroenteritis in humans if ingested in large numbers. Can be isolated from soil and water and is commonly associated with spoilage of food such as eggs, cured meats, fish, and milk and cosmetic.

Thyme “conventional” macerates diminished the growth of L929 fibroblasts cytotoxicity, and “organic” oregano increased the ability of L929 cells to reduce MTT. Cultivation type had shaped influence on the ability of L929 cells to reduce MTT—both organic herbs macerates had significant higher abilities to increase this property; however, conventional cultivated herbs (water macerates) had higher antimicrobial activity and higher polyphenols content. Extracts, isolated from ecological plant cultures, may be used in higher concentrations in formulations for medical use due to their lower cytotoxic activity. This allows achieving a higher concentration of biologically active substances while maintaining biological safety, and faster achievement of the therapeutic and cosmetic effect.

## Figures and Tables

**Figure 1 molecules-26-01047-f001:**
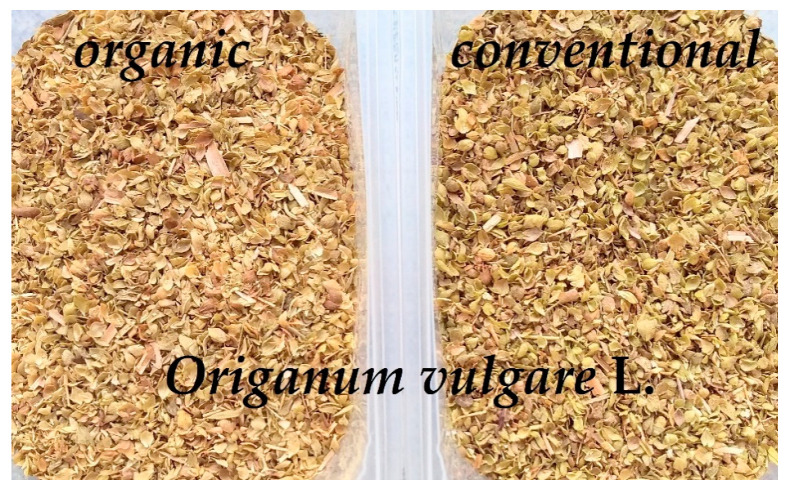
Comparison of colors of dry oregano from organic and conventional cultivation.

**Figure 2 molecules-26-01047-f002:**
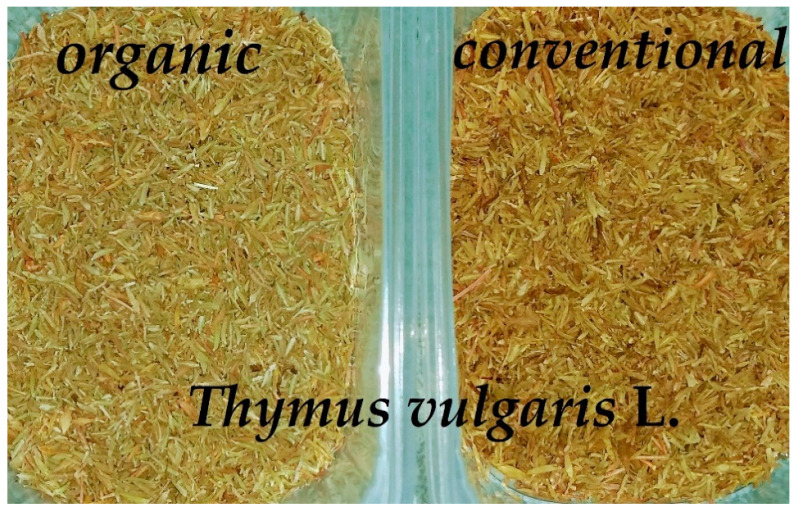
Comparison of colors of dry thyme from organic and conventional cultivation.

**Figure 3 molecules-26-01047-f003:**
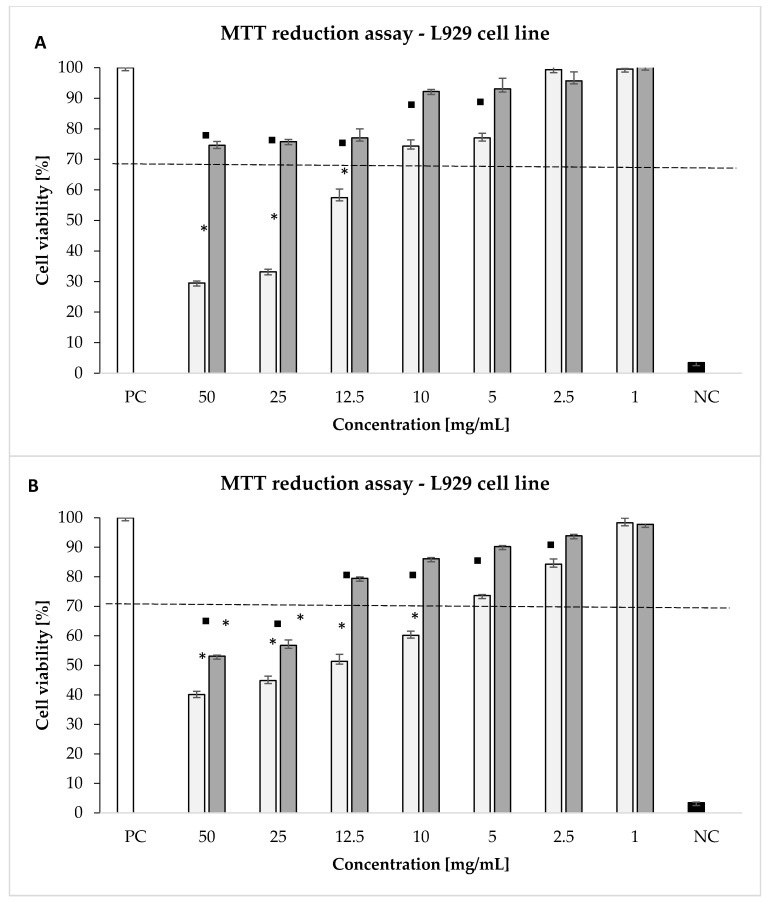
Cytotoxic effect of plants extracts: (**A**) thyme, (**B**) oregano towards L929 cells. The cytotoxicity was assessed by MTT [(3-(4,5-dimethylthiazol-2-yl)-2,5-diphenyltetrazolium bromide)] reduction assay. The cell viability was calculated for four experiments, including three repeats for each compound. Complete RMPI-1640 medium (cRPMI) was used as a positive control (PC) of cell viability (100% viable cells) and 0.03% H_2_O_2_ as a negative control (NC) of cell viability (100% dead inactive cells). Statistical significance: * *p* < 0.05; * untreated cells vs cells treated with tested plants extracts “conventional” extract (light grey bars) vs “organic” extract (dark grey bars).

**Table 1 molecules-26-01047-t001:** The chosen chemical compounds are determined in water macerates of dry oregano and thyme cultivated in conventional (CONV) and organically (ORG).

Herb	Cultivation Type	Polyphenols		Chlorophyll a		Chlorophyll b		Carotenoids	
		mg GAE/mL Sample		μg/1 mL Sample					
oregano	ORG	0.20	a	7.81	cd	12.32	cd	2.07	bcd
	CONV	0.36	bc	4.84	ab	7.21	ab	1.03	abcd
thyme	ORG	0.40	bcd	7.35	cd	11.40	cd	1.67	abc
	CONV	0.42	cd	2.96	ab	4.56	ab	0.76	abc

According to Duncan’s test, the means in columns followed by the same letters are not significantly different at *p* = 0.05.

**Table 2 molecules-26-01047-t002:** Determined values of MIC and MBC for tested water macerates in relation to standard microorganisms.

Microorganisms	Water Macerates				Antimicrobial Reference Standard *	
	Thyme CONV	Thyme ORG	Oregano CONV	Oregano ORG	Gentamicin	Fluconazole
	MIC/MBC (mg/mL)				MIC/MBC (µg/mL)	
**Gram-Positive Bacteria**						
*Staphylococcus aureus* ATCC 25923	12.5/12.5	>50/>50	6.25/6.25	50/50	0.5	nt
*Enterocococcus faecalis* ATCC 29212	25/25	>50/>50	25/25	>50/>50	16	nt
*Bacillus cereus* PCM 1948	6.25/6.25	12.5/12.5	3.125/3.125	6.25/6.25	0.25	nt
*Bacillus subtilis* ATCC 6635	25/25	50/50	6.25/6.25	25/25	0.25	nt
*Listeria monocytogenes* PCM 2191	50/50	50/50	25/25	25/25	0.25	nt
**Gram-Negative Bacteria**						
*Escherichia coli* ATCC 25922	>50/>50	>50/>50	50/50	>50/>50	2	nt
*Pseudomonas aeruginosa* ATCC 27853	0.625/0.625	>50/>50	12.5/12.5	>50/>50	2	nt
*Shigella flexneri* ATCC 12022	50/50	>50/>50	>50/>50	>50/>50	nd ^	nt
*Salmonella* Enteritidis ZMF 279	12.5/12.5	>50	50/50	>50/>50	0.25	nt
**Fungi**						
*Candida albicans* ATCC 10241	>50/>50	>50/>50	>50/>50	>50/>50	nt	5/>5
*Aspergillus brasiliensis* ATCC 16404	>50/>50	>50/>50	>50/>50	>50/>50	nt	5/>5

* Gentamicin—broad-spectrum antibiotic, fluconazole—antifungal chemotherapeutic; ^ not detection; nt—the activity of fluconazole (against bacteria) and gentamicin (against fungi) have not been tested.

## Data Availability

The data presented in this study are available on request from the corresponding author.

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
