# Peer review of "Influence of Agronomic Practice on Total Phenols, Carotenoids, Chlorophylls Content, and Biological Activities in Dry Herbs Water Macerates"

_molecules, 2021, doi:10.3390/molecules26041047_

Round 1
Reviewer 1 Report
Please see comments in attached file.

Author Response
Reviver 1
Authors describes the total phenols and flavonoids content, antimicrobial and cytotoxic activities of thyme and oregano water macerates from ecological and conventional cultivated dry herbs. Generally, the applied methodology is correct. I suggest some corrections before article acceptance:
Please modify title as follow: “Influence of agronomic practice on total phenols and flavonoids content, and biological activities in dry herbs water macerates.
Thank you for your comment, we have changed the title but not directly to suggested one, because we have not determined flavonoids, therefore, the title might be misleading for the readers.
Abstract should be revised in order to highlight the novelty of the information reported in this paper. Please follow this scheme background-methodology applied- results- conclusion. Actually, this section is too descriptive.
Thank you for your comment, we have corrected abstract according to your suggestions.
Materials and Methods section: Please insert samples origin. Where samples are collected? Are bought in the supermarket? Who identify the two herbs? How samples are dried? Please insert information on applied time and temperature.
Our study aimed to evaluate dry herbs spices available on the market, which are used in cooking. Therefore, the analysed herbs were purchased on the market. The manufacturer prepared the form (fragmentation, packaging) and technological processing (harvest, time and form of drying). In order to avoid any discrepancies in the above, both herbs in both types of cultivation were purchased from the same producer.
Results: Please perform correlations analysis to verify if the different content of TPC and TFC influenced the biological activities.
As mentioned in response to question No.1, we have not determined flavonoids (total flavonoids content).
Discussion section: Please insert some information reported in the following articles:
- J Inno Sci Eng 2018, 2(1): 25-33; - Food Chemistry 2009; 117(1):174-180. DOI: 10.1016/j.foodchem.2009.03.095;
- Int J Food Sci Nut. 2015;66(1):50-7. doi: 10.3109/09637486.2014.953454; - Influence of Organic and Conventional Agricultural Practices on Chemical Profile, In Vitro Antioxidant and Anti-Obesity Properties of Zingiber officinale Roscoe. Published: 30 November 2020 by MDPI in The 1st International E-Conference on Antioxidants in Health and Disease session The Biology of Natural Products in Disease Pathophysiology: Mechanisms of Action DOI: 10.3390/CAHD2020-08559.
We have added the suggested literature position.
Conclusion section: Improve this section in order to this query: Ecological vs. conventional: what are the relative impacts on the products?
Thank you for your comment we have fulfilled the conclusion section with the information about the impact of cultivation type.
The manuscript has been subject to professional linguistic verification (attached certificate).
Reviewer 2 Report
Comments for Molecules-1106126
The manuscript focuses on assess if the type of cultivation influenced chosen chemical parameters and antimicrobial activity and shaped cytotoxicity in water macerates of oregano and thyme. The content of the paper is quite valuable, but there are some problem in experimental design should be solved before the manuscript been considered for publication.
Substantial revisions
Q1: L929 cell line is a normal muscle cell. What is the reason for choosing L929 cell line as the test cell viability (MTT assay)? In the Introduction of this article, for 2 kinds of plants (Origanum vulgare L.) Thyme (Thymus vulgaris L.), only food and cosmetic or medicinal materials are mentioned. Does it make sense to use L929 cell normal muscle cells? Does it make sense to use L929 cell normal muscle cells?
Q2: Polyphenols are sensitive for high temperatures; 86 therefore, the drying process reduces their plant tissue content [19]. Please propose to discuss why only the ORG culture group has a lower polyphenols concentration than the CONV culture group during the same drying process in Table 1. The rest of the ingredients are the ORG culture group better than the CONV culture group?
Author Response
Reviewer 2
The manuscript focuses on assess if the type of cultivation influenced chosen chemical parameters and antimicrobial activity and shaped cytotoxicity in water macerates of oregano and thyme. The content of the paper is quite valuable, but there are some problem in experimental design should be solved before the manuscript been considered for publication.
Substantial revisions
Q1: L929 cell line is a normal muscle cell. What is the reason for choosing L929 cell line as the test cell viability (MTT assay)? In the Introduction of this article, for 2 kinds of plants (Origanum vulgare L.) Thyme (Thymus vulgaris L.), only food and cosmetic or medicinal materials are mentioned. Does it make sense to use L929 cell normal muscle cells?Does it make sense to use L929 cell normal muscle cells?
According to ISO 10993-5 (Biological evaluation of medical devices - Part 5: In vitro cytotoxicity tests in Annex C entitled “MTT cytotoxicity test”).
Specific biocompatibility tests and specific issues related to the biocompatibility assessment of medical devices are performed first on the recommended line of mouse fibroblasts (L-929 cells). After excluding the extracts’ cytotoxic effect within the tested concentrations against the L929 line, the tests can be extended to other cell lines.
Therefore, considering the potential use of new biocompound in humans, it is obligatory to exclude its cytotoxic effect in the MTT reduction test performed on the L929 line.
Q2: Polyphenols are sensitive for high temperatures; 86 therefore, the drying process reduces their plant tissue content [19]. Please propose to discuss why only the ORG culture group has a lower polyphenols concentration than the CONV culture group during the same drying process in Table 1. The rest of the ingredients are the ORG culture group better than the CONV culture group?
We have given a possible explanation in the discussion (lines 261-267)
The same problem was considered by Lv at al.: “While some significant differences were observed between organic and conventional herbs in the phenolic profile, a general trend was not seen in favour of either organic or conventional production. This result is in line with some previously published studies(..) but it is in contradiction with others (..), which indicated that organic agriculture had the potential to produce high-quality products with some relevant improvements in terms of contents of antioxidant phytochemicals. The above results indicate a critical need for a large systematic experimental design, with wide variety of food materials grown under different environmental conditions to unambiguously confirm the difference in nutritional quality and benefits of organic and conventionally grown foods”
The manuscript has been subject to professional linguistic verification (attached certificate).
Reviewer 3 Report
Manuscript: molecules-1106126
Title: The comparison of chosen chemical parameters, antimicrobial activity and cytotoxicity of water macerates of thyme and oregano ecological and conventional cultivated dry herbs.
I like the effort of the authors to evaluate the effects of the type of cultivation (ecological/ conventional) on chemical parameters, antimicrobial and cytotoxic activity of water extracts of thyme and oregano.
I would recommend the authors to check carefully the manuscript, since I have detected some mistakes. In the following paragraphs, I will provide clear information to improve the manuscript.
Abstract
Line 25: “Polyphenols content and antimicrobial activity were higher in water macerates obtained from conventional cultivation (independently from herb species) than in pigments in a higher amount in macerates from organic herbs cultivation”. This phrase is a bit confusing, please rephrase.
- Introduction
In my opinion, the first part of the introduction (related to the description of both plants) is well-written and gives important information. However, the last part is a bit disorganized. Have differences caused by the type of crop been observed in other plants? Why is important to assess the differences of the chosen parameters attributed to the type of cultivation? I have detected a few mistakes in this section:
- Line 35: popular plants
- Line 36: of the Mediterranean region
- Line 36: although they became
- Line 38, 78: These sentences should not be separated from the previous text.
- Material & Methods
This section is clear and precise. Instead of using Folin-Ciocalteu method, authors have considered to determine the phenolic content by HPLC?
- Line 133: “The” is not necessary before a species’ name.
- Line 161: In vitro should be in italics.
- Results
The results have been presented in a clear and fluid manner. Tables and Figures present relevant information in a very visual way.
- Discussion
This section is well-written, the authors compared their results with numerous previous studies and they discussed widely the results obtained. However, I would suggest the authors to check this section: there are too long sentences and commas are missing.
- In my opinion, Lines 306-314 do not present relevant information to the discussion.
Conclusions
From my point of view, conclusions are too short and schematic. What is the relevance of this work?
FINAL REMARKS
In my opinion, I am suggesting MAJOR REVISIONS before publishing. The authors have performed an interesting work but they should do some changes to improve its quality.
Author Response
Reviewer 3
Title: The comparison of chosen chemical parameters, antimicrobial activity and cytotoxicity of water macerates of thyme and oregano ecological and conventional cultivated dry herbs.
I like the effort of the authors to evaluate the effects of the type of cultivation (ecological/ conventional) on chemical parameters, antimicrobial and cytotoxic activity of water extracts of thyme and oregano.
I would recommend the authors to check carefully the manuscript, since I have detected some mistakes. In the following paragraphs, I will provide clear information to improve the manuscript.
Abstract
Line 25: “Polyphenols content and antimicrobial activity were higher in water macerates obtained from conventional cultivation (independently from herb species) than in pigments in a higher amount in macerates from organic herbs cultivation”. This phrase is a bit confusing, please rephrase.
Thank you for your comment we have corrected the sentence.
Introduction
In my opinion, the first part of the introduction (related to the description of both plants) is well-written and gives important information. However, the last part is a bit disorganized.
Thank you for your comment. In the last part of the introduction, we described all factors that were/may influence the conducted analysis. Therefore, we were trying to list them one by one.
Have differences caused by the type of crop been observed in other plants?
Type of crop is one of the factors that shapes the mentioned parameters, although we have focused on two plants, cultivated in two ways (and their chemical and microbiological evaluation) cause widening of the factors could make correct and clear comparison impossible.
Why is important to assess the differences of the chosen parameters attributed to the type of cultivation?
In most cases, consumers base their knowledge on basic information, e.g., organic/ecological, are healthy – opposite to conventionally cultivated crops. Also, certain abilities are bounded with analysed herbs (like antimicrobial or free radicals’ scavengers), although these properties are connected with fresh herbs. Because in polish cuisine herbs are used mostly dry as a part of the water solution, we have decided to evaluate and compare these macerate’s chosen properties.
I have detected a few mistakes in this section:
Line 35: popular plants
Line 36: of the Mediterranean region
Line 36: although they became
Line 38, 78: These sentences should not be separated from the previous text.
Thank you very much; we have corrected the sentences.
Material & Methods
This section is clear and precise. Instead of using Folin-Ciocalteu method, authors have considered to determine the phenolic content by HPLC?
Thank you for your comment. The part “polyphenols determination” was conducted as a part of studies program where the students meet different method of determination, in this case, the spectrophotometric one.
Line 133: “The” is not necessary before a species’ name.
Line 161: In vitro should be in italics.
Thank you very much; we have corrected the sentences.
Results
The results have been presented in a clear and fluid manner. Tables and Figures present relevant information in a very visual way.
Thank you for your comment.
Discussion
This section is well-written, the authors compared their results with numerous previous studies and they discussed widely the results obtained. However, I would suggest the authors to check this section: there are too long sentences and commas are missing.
We have corrected the discussion (shortage the sentences and put more attention to correct the interpunction).
In my opinion, Lines 306-314 do not present relevant information to the discussion.
Thank you for your comment- we have corrected or deleted these sentences to be more appropriate to this part of a manuscript.
Conclusions
From my point of view, conclusions are too short and schematic. What is the relevance of this work?
We have fulfilled the conclusion with the basic needed information drawn from the obtained results and summarised our work.
FINAL REMARKS
In my opinion, I am suggesting MAJOR REVISIONS before publishing. The authors have performed an interesting work but they should do some changes to improve its quality.
Thank you for the summary. We have corrected the manuscript in line with the reviewers' comments, and we hope that it has become clearer and more readable and that all doubts and mistakes have been clarified and/or explained either in the text or in the responses to the reviewers.
The manuscript has been subject to professional linguistic verification (attached certificate).
Round 2
Reviewer 2 Report
Comments for Molecules-1106126
The manuscript focuses on assess if the type of cultivation influenced chosen chemical parameters and antimicrobial activity and shaped cytotoxicity in water macerates of oregano and thyme. The content of the paper is quite valuable. In the revised manuscript, My questions in experimental design has been responded by authors. Thus I suggest that the manuscript be considered for publication.
Author Response
Reviewer 2
The manuscript focuses on assess if the type of cultivation influenced chosen chemical parameters and antimicrobial activity and shaped cytotoxicity in water macerates of oregano and thyme. The content of the paper is quite valuable. In the revised manuscript, My questions in experimental design has been responded by authors. Thus I suggest that the manuscript be considered for publication.
Thank you for your comment.
Reviewer 3 Report
Authors have taken the previous comments into consideration and the text has been improved. However, I have little comments for them.
Introduction:
In the paragraph “Technological processes such as drying increase the herbs…”, I think that a phrase like “Different factors have been demonstrated to influence the content of the biologically active compounds, such as technological process, cultivation method, climatic conditions or variety …” could help to better organize this part of the introduction. Then, the authors can describe the different variables mentioned.
Results:
In Figure 3, the letter A is missing in the first graphic.
Discussion:
“Chlorophyll a and b differ in a side chain’s composition…”: and shouldn’t be in italics.
“herbs cultivated in organic fertilizers by the possible improvement of nutrient accessibility by organically fertilisers and…”, please, rephrase.
“effective against the strains of P. aeruginasa”: P. aeruginosa.
“thyme extract against B.subtilis”: B. subtilis (a space is missing).
“Our study showed that extract isolated from thyme “organic” does not show cytotoxic activity than the extract isolated from thyme “conventional”. Please, rephrase.
“They showed that the T. vulgaris extract at 25, 50…”: T. vulgaris should be in italics.
Why authors have included the study of Loizzo et al. (2009) in the discussion of cytotoxic results? The study evaluated the anti-inflammatory properties of O. ehrenbergii and O. syriacum in RAW 264.7.
Conclusion:
Authors have further elaborated their conclusions. In my opinion, the last paragraph should be reduced a bit and used as first part of the conclusions, since the authors have stated the importance of the work: the cultivation method influences the composition and activities of the water extracts of the plants.
“activity against the strains of P. aeruginosa…”: P. aeruginosa.
Author Response
Reviewer 3
Authors have taken the previous comments into consideration and the text has been improved. However, I have little comments for them.
Introduction:
In the paragraph “Technological processes such as drying increase the herbs…”, I think that a phrase like “Different factors have been demonstrated to influence the content of the biologically active compounds, such as technological process, cultivation method, climatic conditions or variety …” could help to better organize this part of the introduction. Then, the authors can describe the different variables mentioned.
Thank you for your suggestion we have add the sentence and merged it with drying process.
Results:
In Figure 3, the letter A is missing in the first graphic.
Thank you we corrected the mistake.
Discussion:
“Chlorophyll a and b differ in a side chain’s composition…”: and shouldn’t be in italics.
Thank you for your suggestion we have correct the italics.
“herbs cultivated in organic fertilizers by the possible improvement of nutrient accessibility by organically fertilisers and…”, please, rephrase.
“Our study showed that extract isolated from thyme “organic” does not show cytotoxic activity than the extract isolated from thyme “conventional”. Please, rephrase.
Thank you for your suggestion we have rephrased the sentences, into:
“Onofrei et al. (2017) explained the higher content of pigments in herbs cultivated with organic fertilizers by the possible improvement in the availability of nutrients caused by these fertilizers, and thus enhancing the metabolic pathways (e.g., photosynthesis) synthesis of several plant secondary compounds [36].” And,
“Our study showed that extract isolated from thyme “organic” does not show cytotoxic activity according to ISO 10993‐5 in all concentrations used in this study (1-50 mg/ml) whereas the activity of extract isolated from thyme “conventional” was below the cy-totoxicity norm in the range 12,5-50 mg/ml”
“effective against the strains of P. aeruginasa”: P. aeruginosa.
Thank you for your suggestion we have correct the latin name.
“thyme extract against B.subtilis”: B. subtilis (a space is missing).
Thank you for your suggestion we have add a space.
“They showed that the T. vulgaris extract at 25, 50…”: T. vulgaris should be in italics.
Thank you for your suggestion we have correct the latin name.
Why authors have included the study of Loizzo et al. (2009) in the discussion of cytotoxic results? The study evaluated the anti-inflammatory properties of O. ehrenbergii and O. syriacum in RAW 264.7.
That was a suggestion of Rewiever 1. The context of this citation, which was recommended by the Rewiever 1 in the round 1st has been extended:
“New plant extracts are also tested for antioxidant activity in conjunction with the modulation of the inflammatory response. For example, Loizzo et al. (2009) examined different oregano’s oils varieties inhibition of NO production in the murine monocytic macrophage cell line RAW 264.7, obtaining an IC50 value of 66.4 μg/ml and >200 μg/ml for O. ehrenbergii and O. syriacum, respectively [53]. The antioxidant activity of our preparations with the use of gastric epithelial cells is planned to be tested in the context of minimizing the risk of epithelial damage and the development of an inflammatory reaction”.
In the context of the anti-cancer activity of plant extracts, we have added the following sentence: “Plant preparations with a cytotoxic effect against cancer cells are extremely valuable”.
Conclusion:
Authors have further elaborated their conclusions. In my opinion, the last paragraph should be reduced a bit and used as first part of the conclusions, since the authors have stated the importance of the work: the cultivation method influences the composition and activities of the water extracts of the plants.
“activity against the strains of P. aeruginosa…”: P. aeruginosa.
Thank you for your suggestions, we have reduced and improve the conclusion according to your tips.
This manuscript is a resubmission of an earlier submission. The following is a list of the peer review reports and author responses from that submission.